# Learning to Assimilate in Chaotic Dynamical Systems

**Michael McCabe**
Department of Computer Science
University of Colorado Boulder
michael.mccabe@colorado.edu

**Jed Brown**
Department of Computer Science
University of Colorado Boulder
jed@jedbrown.org

## Abstract

The accuracy of simulation-based forecasting in chaotic systems is heavily dependent on high-quality estimates of the system state at the time the forecast is initialized. Data assimilation methods are used to infer these initial conditions by systematically combining noisy, incomplete observations and numerical models of system dynamics to produce effective estimation schemes. We introduce *amortized assimilation*, a framework for learning to assimilate in dynamical systems from sequences of noisy observations with no need for ground truth data. We motivate the framework by extending powerful results from self-supervised denoising to the dynamical systems setting through the use of differentiable simulation. Experimental results across several benchmark systems highlight the improved effectiveness of our approach over widely-used data assimilation methods.

## 1 Introduction

Forecasting in the geosciences is an initial value problem of enormous practical significance. In high value domains like numerical weather prediction [1], climate modeling [2], atmospheric chemistry [3], seismology [4], and others [5–7], forecasts are produced by estimating the current state of the dynamical system of interest and integrating that state forward in time using numerical models based on the discretization of differential equations. These numerical models are derived from physical principles and possess desirable extrapolation and convergence properties. Yet despite the efficacy of these models and the vast amount of compute power used in generating these forecasts, obtaining highly accurate predictions is non-trivial. Discretization introduces numerical errors and it is often too computationally expensive to directly simulate the system of interest at the resolution necessary to capture all relevant features.

Further complicating matters in geoscience applications is that many systems of interest are chaotic, meaning that small errors in the initial condition estimates can lead to significant forecasting errors over relatively small time frame [8]. This can be problematic when the initial condition for a forecast is current state of the Earth, a large-scale actively evolving system that cannot be directly controlled. Modern numerical weather prediction (NWP) models utilize hundreds of millions of state variables scattered over a three-dimensional discretization of the Earth's atmosphere [9]. To perfectly initialize a simulation, one would need to know the true value of all of these state variables simultaneously. This data is simply not available. In reality, what is available are noisy, partial measurements scattered non-uniformly in time and space.

The inverse problem of estimating the true state of a dynamical system from imperfect observations is the target of our work and of the broader field known as data assimilation. Data assimilation techniques produce state estimates by systematically combining noisy observations with the numerical model [10, 9]. The data assimilation problem differs from other state estimation problems in that the numerical model acts both as an evolution equation and as a mechanism for transporting information from regions where observations are dense to regions where observations are sparse [11]. Thus for accurate estimates of the full system state, it is important to utilize both the model and observations.

35th Conference on Neural Information Processing Systems (NeurIPS 2021).

One recent trend in deep learning research has been the push to develop neural network models to replace expensive, oft repeated processes. This incurs a large upfront cost to train the network in exchange for a faster solution to subsequent iterations of the problem. These amortized methods were initially developed for statistical inference [12] but have also been used in the context of numerical simulation [13], and meta-learning [14]. Our work extends this approach to variational data assimilation [15].

In this work, we introduce a self-supervised framework for learning to assimilate which we call amortized assimilation. Here, we use the term self-supervised to indicate that our method learns to assimilate entirely from trajectories of noisy observations without the use of ground truth data during training. Our design incorporates the objective flexibility of variational assimilation but amortizes the expense of solving the nonlinear optimization problem inherent to variational methods into the training of a neural network that then behaves as a sequential filter.

Our contributions are both theoretical and empirical. In Section 3.1, we introduce an amortized assimilation architecture based on the Ensemble Kalman Filter [16]. We then develop the theory of amortized assimilation in Section 3.2 by extending the powerful self-supervised denoising results of Batson and Royer [17] to the dynamical systems setting through the use of differentiable simulation. We then support this theory through a set of numerical experiments[1] in Section 5, where we see that amortized assimilation methods match or outperform conventional approaches across several benchmark systems with especially strong performance at smaller ensemble sizes.

## 2 Preliminaries

**Problem Setting**   In this work, we consider a dynamical system with state variable $\mathbf{x}(t) \in C \subset \mathbb{R}^d$ where $C$ is a compact subset of $\mathbb{R}^d$ and system dynamics defined by the differential equation:

$$\frac{d\mathbf{x}}{dt} = g(\mathbf{x}(t)) \tag{1}$$

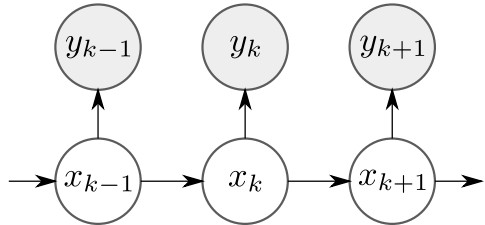

Figure 1: Observations are assumed to be generated from a Hidden Markov Model.

with $g$ as a time-invariant, Lipschitz continuous map from $\mathbb{R}^d \to \mathbb{R}^d$. The state is observed at a sequence of $K$ discrete time points $\{\tau_0, \tau_1, \ldots, \tau_K\}$ generating the time-series $\mathbf{y}_{0:K} = \{\mathbf{y}_0, \mathbf{y}_1, \ldots, \mathbf{y}_K\}$ where $0 \leq k \leq K$ is an index corresponding to the time point $\tau_k$. These observations are imperfect representations of the system states $\mathbf{x}_{0:K} = \{\mathbf{x}_0, \mathbf{x}_1, \ldots \mathbf{x}_K\}$ generated from observation operators of the form:

$$\mathbf{y}_k = \mathcal{H}_k(\mathbf{x}_k) + \boldsymbol{\eta}_k \tag{2}$$

where $\mathcal{H}_k$ is an arbitrary potentially nonlinear function and $\boldsymbol{\eta}_k \sim \mathcal{N}(0, \sigma_k^2 \mathbf{I})$. This noise model is often referred to as a Hidden Markov Model (Figure 1). In the classical assimilation setting, the observation operators are assumed to be known *a priori* though our proposed method is capable of learning a subset of observation operators. For a problem like numerical weather prediction, these operators may represent measurements taken of certain atmospheric quantities at a particular measurement location.

Data assimilation is then the inverse problem of estimating the true trajectory $\mathbf{x}_{0:K}$ from our noisy observations $\mathbf{y}_{0:K}$ and our model of the system evolution:

$$\mathbf{x}(\tau_{k+1}) = \mathcal{M}_{k:k+1}(\mathbf{x}(\tau_k)) = \mathbf{x}(\tau_k) + \int_{\tau_k}^{\tau_{k+1}} g(\mathbf{x}(t)) \, dt. \tag{3}$$

**Sequential Filtering**   In the absence of ground truth, it is practical to present data assimilation from a statistical perspective. In particular, the quantity of interest that we are concerned with in this paper is the *filtering distribution* $p(\mathbf{x}_k \mid \mathbf{y}_{0:k})$. Efficient algorithms for estimating the filtering distribution

---

[1]Code available at https://github.com/mikemccabe210/amortizedassimilation.

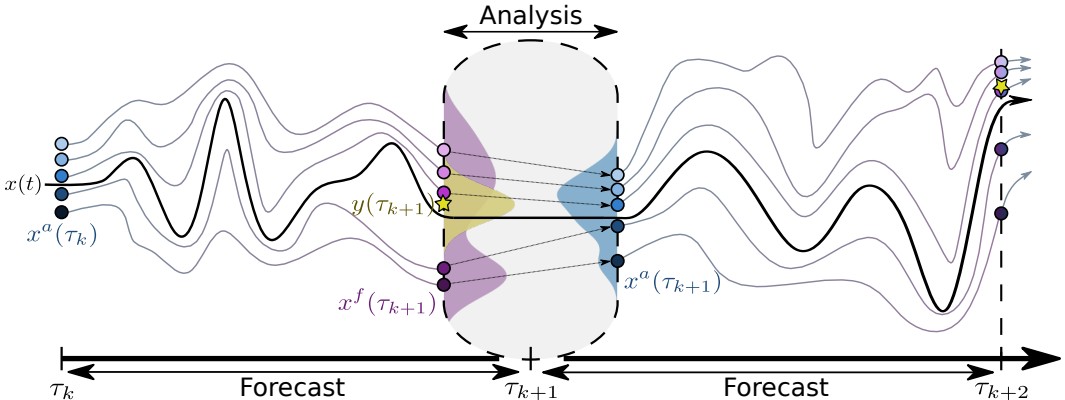

Figure 2: Ensemble filters model the time-evolution of uncertainty under nonlinear dynamics by maintaining a set of samples from the state distribution and simulating their trajectories forward in time. Observations are then used to refine these forecast estimates in what is called the analysis step.

are derived from exploiting dependence relationships in the generative model to produce the two step cycle:

$$\textbf{Forecast:} \qquad p(\mathbf{x}_k \mid \mathbf{y}_{0:k-1}) = \int_{\mathbb{R}^d} p(\mathbf{x}_k \mid \mathbf{x}_{k-1}) \, d\mathbf{x}_{k-1} \qquad (4a)$$

$$\textbf{Analysis:} \qquad p(\mathbf{x}_k \mid \mathbf{y}_{0:k}) \propto p(\mathbf{y}_k \mid \mathbf{x}_k) p(\mathbf{x}_k \mid \mathbf{y}_{0:k-1}) \qquad (4b)$$

When evaluating data assimilation methods, the objective function is often evaluated with respect to the "analysis" estimates as these act as these are the initial conditions used for the next forecast phase. In subsequent sections, we will use $\mathbf{x}_k^f$ and $\mathbf{x}_k^a$ to refer to forecast and analysis estimates of $\mathbf{x}_k$ respectively.

In the case of linear dynamics with Gaussian uncertainties, the Kalman filter [18] provides both the forecast and analysis distributions exactly in closed form. However, the general case is significantly more complicated. Under nonlinear dynamics, computing the forecast distribution requires the solution of the Fokker-Planck partial differential equation [19]. While finding an exact solution to these equations is computationally infeasible, one can produce what is called an ensemble estimate of the forecast distribution using Sequential Monte Carlo methods [20].

Ensemble filters (Figure 2) replace an exact representation of the uncertainty over states with an empirical approximation in the form of a small number of samples called particles or ensemble members. The forecast distribution can then be estimated by integrating the dynamics forward in time for each ensemble member independently. Allowing for full generality leads to particle filter approaches while enforcing Gaussian assumptions leads to the Ensemble Kalman filter (EnKF) [16]. The former is rarely used in data assimilation settings as particle filters can become degenerate in high dimensions [21] while the latter has been quite successful in practice. Like the classical Kalman filter, the EnKF assumes that both the state distribution and observation likelihoods are Gaussian. The latter is often reasonable, but the former is a rather strong assumption under nonlinear dynamics.

Ensemble methods possess an inherent trade-off in that larger ensembles more accurately model uncertainty, but each ensemble member requires independently integrating the dynamics forward in time which can be significantly more expensive than the assimilation step itself. Bayesian filters rely on robust covariance estimates to maintain stability which can be difficult to obtain when using a small ensemble. Heuristics such as covariance inflation [22] and localization [23] have been developed to improve stability at smaller ensemble sizes, but there remains value in developing methods that improve upon the accuracy of these approaches.

**Variational Assimilation**   The alternative to sequential filters is variational assimilation. Variational assimilation methods, like 4D-Var [24], directly solve for the initial conditions of a system by finding the state $\mathbf{x}_0$ which minimizes the negative log posterior density through nonlinear optimization. This is expensive, but gradient-based optimization utilizing the adjoint of the numerical model avoids the persistent Gaussian state assumptions necessary for tractable Bayesian filters. Furthermore, these

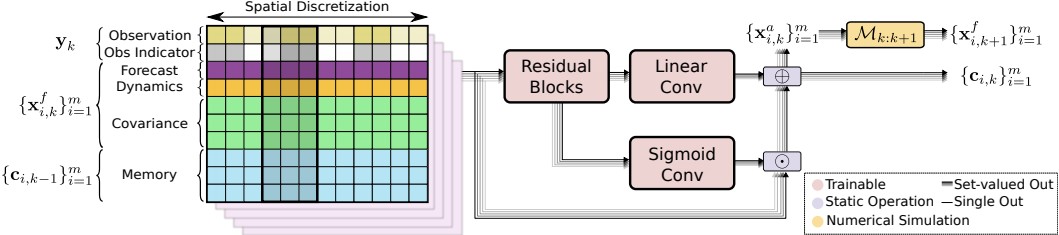

Figure 3: The AmEnF replaces the analysis step of a conventional ensemble filter with a neural network trained using historical data. The model uses an augmented state which includes both standard inputs as well as recurrent memory.

methods can be augmented by auxiliary objectives promoting concordance with physical properties like conservation laws.

# 3 Amortized Assimilation

## 3.1 Amortized Ensemble Filters

Our proposed approach attempts to combine the flexibility of variational methods with the efficiency of sequential filters by training a neural network to act as an analysis update mechanism. Before diving into the theory, we present the general amortized assimilation framework from the perspective of a specific architecture which we call the Amortized Ensemble Filter (AmEnF). The AmEnF (Figure 3) replaces the EnKF analysis equations with a parameterized function in the form of a neural network whose weights are optimized for a specific dynamical system during training. By unrolling a fixed number of assimilation cycles as first proposed in Haarnoja et al. [25], the AmEnF can be trained efficiently using backpropagation through time [26]. The loss used for this training will be discussed in more detail in Section 3.2.

One advantage of decoupling from the statistical model in this way is that it allows us to incorporate more information into our analysis process. Like the EnKF, the analysis ensemble members are computed using the forecast values $\mathbf{x}_{i,k}^f$ for each ensemble member $i \in [m]$ with $m$ denoting the number of ensemble members, the ensemble covariance $P_k^f$, and the observations $\mathbf{y}_k$. However, we augment these features with the dynamics $g(\mathbf{x}_{i,k}^f)$ and coordinate-wise recurrent memory $\mathbf{c}_{i,k-1}$ resulting in the following update equations:

$$\mathbf{x}_{i,k}^f = \mathcal{M}_{k-1:k}(\mathbf{x}_{i,k-1}^a) \qquad P_k^f = Cov(\{\mathbf{x}_{i,k}^f\}_{k=1}^m)$$
$$\mathbf{z}_{x_i}, \mathbf{z}_{c_i} = f_L(\mathbf{x}_{i,k}^f, P_k^f, \mathbf{y}_k, g(\mathbf{x}_{i,k}^f), \mathbf{c}_{i,k-1}) \qquad \boldsymbol{\lambda}_{x_i}, \boldsymbol{\lambda}_{c_i} = f_N(\mathbf{x}_{i,k}^f, P_k^f, \mathbf{y}_k, g(\mathbf{x}_{i,k}^f), \mathbf{c}_{i,k-1}) \quad (5)$$
$$\mathbf{x}_{i,k}^a = \boldsymbol{\lambda}_{x_i} \odot \mathbf{x}_{i,k}^f + \mathbf{z}_{x_i} \qquad \mathbf{c}_{i,k} = \boldsymbol{\lambda}_{c_i} \odot \mathbf{c}_{i,k-1} + \mathbf{z}_{c_i}$$

where $f_L$ and $f_N$ are neural networks with linear and sigmoid final activations respectively. We implement these as a single network which is split only at the final layer. We observed notable gains to training stability through the inclusion of this sigmoid-gated final layer. For scalability, we do not use the full covariance matrix as an input to the model. Rather, we include local covariance entries as additional feature channels where each channel represents the covariance with the state value at a fixed relative spatial position. For instance in a 1D system, the relative covariance channels may contain the variance at the given point and the covariances with the points to the left and right.

The inclusion of recurrent memory may seem out of place when the dynamics are assumed to be Markovian. However, while the dynamics themselves are Markovian, the data assimilation process is not. The inclusion of memory allows the model to learn from the past without explicitly including previous states and observations. This is a significant benefit in practice. The other non-standard inclusion, the local dynamics, led to a small performance boost as well. As our method is not reliant on generative models of the inputs, additional features could be added trivially.

## 3.2 Self-Supervised Assimilation

The major obstacle to training an assimilation model is the lack of ground truth data. In this section, we derive theoretical motivation for the self-supervised training procedure that lies at the heart of amortized assimilation. Recent work in image denoising has shown that self-supervised methods are a promising alternative to denoising methods with explicit noise models. Lehtinen et al. [27] initially demonstrated the ability to denoise images without clean ground truth images by using multiple noise samples. Batson and Royer [17] expanded this idea to the more general $\mathcal{J}$-invariance framework which exploits independence assumptions between noise in different partitions of the state vector to derive a valuable decomposition of the denoising loss. Techniques developed using this approach have exhibited comparable performance to explicitly supervised denoising across multiple applications [28–30]. In our work, we extend this approach to the dynamical systems setting using differentiable simulation. As we do not need the full generality of the $\mathcal{J}$-invariance framework to derive our results, we begin by restricting Proposition 1 from Batson and Royer [17] to our setting.

**Lemma 1** (*Noise2Self – Restricted*). *Suppose concatenated noisy observation vector $\boldsymbol{y} = [\boldsymbol{y}_k; \boldsymbol{y}_{k+1}]$ is an unbiased estimator of concatenated state vector $\boldsymbol{x} = [\boldsymbol{x}_k; \boldsymbol{x}_{k+1}]$ and that the noise in $\boldsymbol{y}_k$ is independent from the noise in $\boldsymbol{y}_{k+1}$. Now let $\boldsymbol{z} = f(\boldsymbol{y}) = [\boldsymbol{z}_k; \boldsymbol{z}_{k+1}]$. If $f$ is a function such that $\boldsymbol{z}_{k+1}$ does not depend on the value of $\boldsymbol{y}_{k+1}$ then:*

$$\mathbb{E}_{\boldsymbol{y}}\|f(\boldsymbol{y})_{k+1} - \boldsymbol{y}_{k+1}\|^2 = \mathbb{E}_{\boldsymbol{y}}[\|f(\boldsymbol{y})_{k+1} - \boldsymbol{x}_{k+1}\|^2 + \|\boldsymbol{y}_{k+1} - \boldsymbol{x}_{k+1}\|^2] \tag{6}$$

While the proof of this lemma can be found in the supplementary materials (C.1), the main takeaway is that if we are able to define a search space for our denoising function such that all $f$ in the space have the stated independence property, then the expected self-supervised denoising loss can be decomposed into the expected supervised loss and the noise variance. As the noise variance term is irreducible, this decomposition implies that minimizing the expected self-supervised loss is equivalent to minimizing the expected supervised loss.

Moving back over to data assimilation, our ultimate goal is to train a neural network to act as our denoising agent. Letting $f_\theta$ denote a neural network parameterized by weights $\theta$, we can formalize the objective that we'd actually like to minimize as the supervised analysis loss:

$$\mathcal{L}^a(\theta) = \frac{1}{K-1} \sum_{k=1}^{K-1} \left\| \left( \frac{1}{m} \sum_{i=1}^{m} f_\theta(\mathbf{x}_{i,k}^f, P_k^f, \mathbf{y}_k, g(\mathbf{x}_{i,k}^f), \mathbf{c}_{i,k-1}) \right) - \mathbf{x}_k \right\|^2. \tag{7}$$

This supervised loss reflects that our goal is to produce the best initial condition estimates at the start of a forecast window. Unfortunately, $\mathbf{x}_k$ is unknown so this objective cannot be used to train a denoising model. In the amortized assimilation framework, we instead minimize what we call the self-supervised forecast loss:

$$\mathcal{L}^{ssf}(\theta) = \frac{1}{K-1} \sum_{k=1}^{K-1} \left\| \left( \frac{1}{m} \sum_{i=1}^{m} \mathcal{H}_{k+1}(\mathcal{M}_{k:k+1} \circ f_\theta(\mathbf{x}_{i,k}^f, \cdot)) \right) - \mathbf{y}_{k+1} \right\|^2. \tag{8}$$

The self-supervised approach (Figure 4) has the obvious advantage that it uses readily available noisy observations as the training target. However, we can also show that it is theoretically well-motivated under Lemma 1. Differentiable simulation acts as a bridge between our denoised estimate and future observations which are assumed to have independently sampled noise under the generative model described in Section 2. While Lemma 1 is stated in terms of one step ahead forecasting, similar to prior work [31], we observe significantly improved performance by unrolling the procedure over multiple steps.

The power of this approach lies in the fact that the forecast objective can actually be used to bound the analysis objective so that minimizing the forecast objective is equivalent to minimmizing an upper bound on the analysis objective. To show this, apart from the previously specified generative model assumptions, we rely on two additional assumptions:

**Assumption 1** (Uniqueness of Initial Value Problem (IVP) Solution). *The autonomous system evolution equation $x'(t) = g(x(t))$ is deterministic and L-Lipschitz continuous in $x$ thus admitting a unique solution to the initial value problem under the Picard-Lindelöf theorem [32].*

**Assumption 2** (Unbiased Observation Operators). *For all observation operators $\mathcal{H}_k$, the observation $y_k$ is an unbiased estimator of some subset of $x_k$, ie. if $x_k^S$ is the restriction of $x_k$ to some subset of features $S$ then $\mathbb{E}[y_k \mid x_k^S] = x_k^S$.*

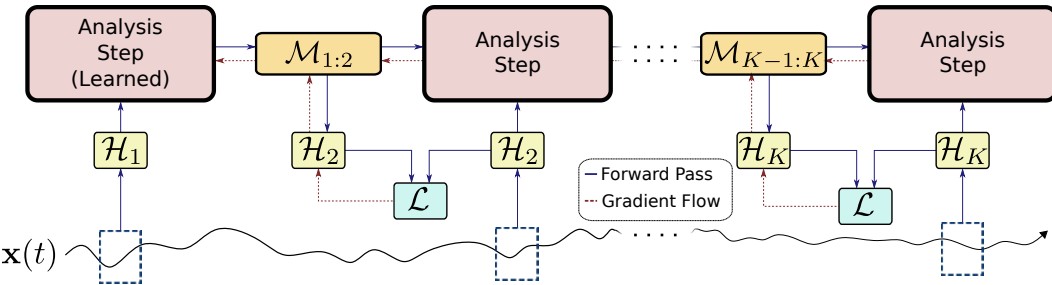

Figure 4: The self-supervised training process is unrolled across multiple filtering steps. Loss is evaluated between "pseudo-observations" and noisy observations obtained from the system.

We also define an additional loss, the supervised forecast loss $\mathcal{L}^f$ as the mean squared error between the forecast state and the true state. For the purposes of our analysis, we'll assume that $S$ is the full state such that $\mathbb{E}[y_k \mid x_k] = x_k$ though this will not be the case in our numerical experiments. Our first result focuses on the case where the supervised loss is driven to zero.

**Proposition 1** (Zero Loss Regime). *Under the stated assumptions, let $f_\theta$ denote a family of functions parameterized by $\theta$ for which $\min_\theta \mathcal{L}^f(\theta) = 0$. For any $\theta$ which achieves this minimum, it is also true that $\mathcal{L}^a(\theta) = 0$ and that $\theta$ is in set of minimizers of $\mathbb{E}_{\mathbf{y}_{0:K}}[\mathcal{L}^{ssf}(\theta)]$.*

The proof can be found in the supplementary materials (C.2). This case provides important motivation by showing that if a perfect denoiser exists and is contained within our search space then it is also in the set of minimizers for our expected self-supervised objective. This does not necessarily mean that such a function will be discovered as the objective is non-convex and in practice, we're minimizing the self-supervised loss over a finite sample rather than over the expectation of the noise distribution.

For the more general case, with sufficiently smooth dynamics, one might expect that over short time horizons, a function which minimizes the forecast loss should perform well for the analysis loss, this is not guaranteed and depends on the properties of the system under study. For our bound, we only consider the supervised losses as per the decomposition in Lemma 1, minimizing the expected self-supervised forecast loss is equivalent to minimizing the expected supervised forecast loss.

**Proposition 2** (Non-zero Loss Regime). *Under the previously stated assumptions, the supervised analysis loss can be bounded by the supervised forecast loss as:*

$$\frac{1}{K-1} \sum_{k=1}^{K-1} \left\| \left( \frac{1}{m} \sum_{i=1}^{m} f_\theta(\cdot) \right) - \boldsymbol{x}_k \right\| \leq \frac{e^{L(\tau_{k+1}-\tau_k)}}{K-1} \sum_{k=1}^{K-1} \max_{i \in [m]} \left\| \mathcal{M}_{k:k+1} \circ f_\theta(\boldsymbol{x}_{i,k}^f, \cdot) - \boldsymbol{x}_k \right\| \quad (9)$$

The proof (found in C.3) simply uses the Lipschitz coefficient to bound the system Lyapunov exponents which govern the evolution of perturbations. While the tightness of the bound is system specific, large Lyapunov exponents can be offset by more frequent assimilation cycles. In Section 5, our numerical results suggest that training under this loss is effective in practice for a number of common test systems designed to simulate real-world phenomena.

### 3.3 Implementation Details

**Avoiding Ensemble Collapse** Ensemble-based uncertainty estimates are an elegant solution to the problem of computing the evolution of uncertainty under nonlinear dynamics, but without assumptions on the noise model, evaluating the analysis uncertainty is non-trivial. A well-known problem in deep learning-based uncertainty quantification is feature collapse [33, 34], a phenomenon in which the hidden representations resulting from regions of the input space are pushed arbitrarily closely together by the neural network. In our ensemble-based uncertainty quantification scheme, this poses an issue as ensemble members can quickly converge over a small number of assimilation steps resulting in ensemble collapse.

We address this by combining ensemble estimation with MC Dropout [35] which interprets a pass through a dropout-regularized neural network as a sample from a variational distribution over network weights. Thus, for standard usage of MC Dropout one can estimate uncertainty in the predictive

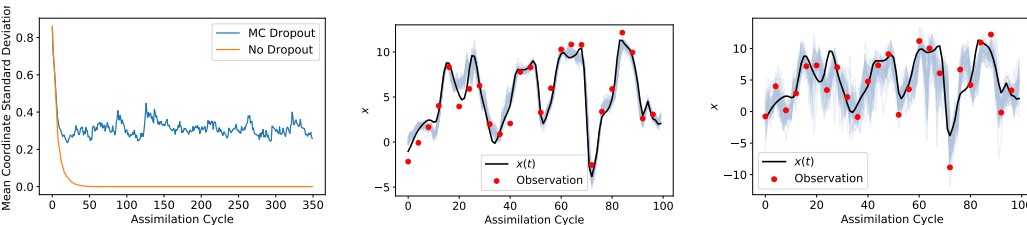

(a) Mean ensemble spread over as-
similation cycles

(b) Ensemble trajectories with ob-
servation noise $\sigma = 1.0$

(c) Ensemble trajectories with obser-
vation noise $\sigma = 2.5$

Figure 5: MC Dropout introduces an additional stochastic element that stabilizes the ensemble resulting in realistic relationships between the ensemble spread and system uncertainty.

distribution by using multiple dropout passes to integrate out weight uncertainty. In our case, we further have to account for uncertainty over the input distribution which is represented by our forecast ensemble. We employ a relatively simple Monte Carlo scheme in which the forecast ensemble members are assimilated by a single pass through a dropout-regularized network with independent dropout masks giving us $m$ samples from the predictive distribution as our analysis ensemble. While this is a crude approximation, we found it resulted in more consistent performance with less tuning compared to the sensitivity constraints found in recent uncertainty quantification methods [33, 36, 37].

**Incomplete Observations** One important concern in data assimilation is handling a variety of observation operators as the full state will rarely be observed at one time point. We discuss two processes for addressing this in amortized filters. The first, which is far more flexible but less effective at exploiting prior knowledge, is to initialize an network with independent weights corresponding to each observation type. This can be used with any observation type, even ones with unknown relationships to the system state. These unknown observation operators can be trained by evaluating the loss at points with future, known observation operators and passing the gradient signal back in time through the differentiable simulation connecting the time points.

The second, which we implement in the AmEnF, is to use our knowledge of the spatial distribution of the observation operators to assign the observed values corresponding to certain locations as channels in the input representation. Coordinates which are not observed can be masked and an additional channel is appending indicating whether a particular coordinate was observed during the given assimilation cycle. This is depicted by the shading in the observation channels of Figure 3. More details on the masking process can be found in the supplementary material.

## 4 Related Work

Work on the intersection of deep learning and data assimilation has exploded in recent years. Some early efforts explicitly used reanalysis data produced by traditional assimilation methods as a target for supervised training [38, 39]. These approaches are limited as they cannot reasonably expect to outperform the method used to produce the reanalysis data in terms of accuracy. More recently, significant effort has gone into using traditional assimilation alongside learned surrogate models to stabilize training, learn augmented dynamics, or accelerate simulation with larger ensembles [40–43].

The work most comparable to our own is that which uses machine learning methods to learn a component of the assimilation mechanism directly in continuous nonlinear systems [44, 45]. Our method differs from Grönquist et al. [44] in that we use differentiable simulation to enable self-supervised learning. While Frerix et al. [45] incorporates differentiable simulation, their Learned Inverse Operator framework acts as a component of the 4D-Var algorithm and still requires solving an expensive optimization problem at each assimilation step rather than using using a closed form update. Outside of the field of data assimilation specifically, there has also been significant work into neural parameterization of Bayesian filters [25, 46–54].

# 5 Experiments

We examine the empirical performance of amortized assimilation from two perspectives. In the first, we derive empirical justification for our design choices, comparing the performance of the various objective functions and qualitatively examining the evolution of uncertainty in the ensemble.

The second objective is to compare the performance of our amortized model, the AmEnF, against standard data assimilation methods over multiple common benchmark dynamical systems. All methods tested in this section have explicit knowledge of the system dynamics. Due to the challenge of learning chaotic dynamics, we found that methods which rely on learning the dynamics were uncompetitive.

For all experiments, we generate a training set consisting of 6000 sequences consisting of 40 assimilation steps each. The validation set consists of a single sequence of an additional 1000 steps and the test set is a further 10,000 steps. AmEnF models are developed in PyTorch [55] using the torchdiffeq [56] library for ODE integration. Models are trained on a single GTX 1070 GPU for 500 epochs using the Adam [12] optimizer with initial learning rate $8e - 4$ with a warm-up over 50 iterations followed by halving the learning rate every 200 iterations. All experiments are repeated over ten independent noise samples and error bars indicate a single standard deviation.

## 5.1 Qualitative Evaluation

Experiments in this section are performed using the Lorenz 96 system [57]. Lorenz 96 is a system of coupled differential equations intended to emulate the evolution of a single atmospheric quantity across a latitude circle.

$$\dot{x}_j = (x_{j+1} - x_{j-2})x_{j-1} - x_j + F \tag{10}$$

The system is defined to have periodic boundary conditions $x_{D+1} = x_1$, $x_0 = x_D$, and $x_{-1} = x_{D-1}$ where $D$ is the number of system dimensions. We set the number of dimensions to 40 and the forcing value to $F = 8$. This is perhaps the most widely used test system in data assimilation and is what we will use the explore the behavior of the AmEnF model. Unless explicitly stated otherwise, all experiments in this section were performed with $\sigma$=1 and $m = 10$.

**Ensemble Behavior** Figure 5 demonstrates the benefit of incorporating MC Dropout. In the left figure, we see that the ensemble variance quickly collapses to zero using deterministic networks while it varies along a stable range using MC Dropout. More interesting are the central and right figures which compare the ensemble member trajectories (light blue) with the true trajectory (black) for a single spatial coordinate across several noise settings. We can see that the uncertainty patterns in our ensemble behave largely as one would hope. Less certainty in the system results in a larger ensemble spread.

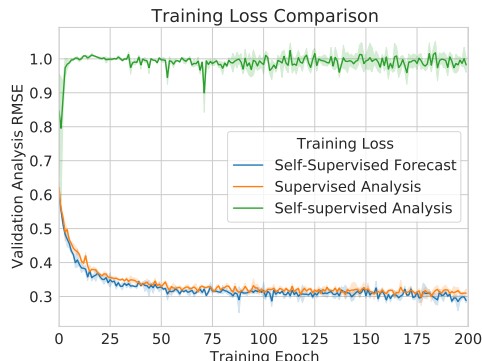

Figure 6: Analysis RMSE evaluated on the validation set across models trained using different losses for $\sigma = 1$.

Of particular note is the behavior near outlier observations. We can see that while the ensemble trajectories occasionally approach such observations, when the ensemble spread is tight the learned assimilator trusts the current forecast trajectory and the ensemble members do not move onto a worse path to match the observation.

**Training Objective** Figure 6 shows a comparison between the test-time supervised analysis loss across AmEnF models trained using several objective functions. The problem used for comparison is the fully observed Lorenz 96 system with observation noise $\sigma = 1$. As one might anticipate, the self-supervised analysis objective, the only objective available without differentiable simulation, results in the AmEnF learning an identity mapping from the observation so that the error is exactly the observation variance. The supervised analysis (which we are able to use since this is simulated data) and self-supervised forecast training approaches result in learning a legitimate denoising function.

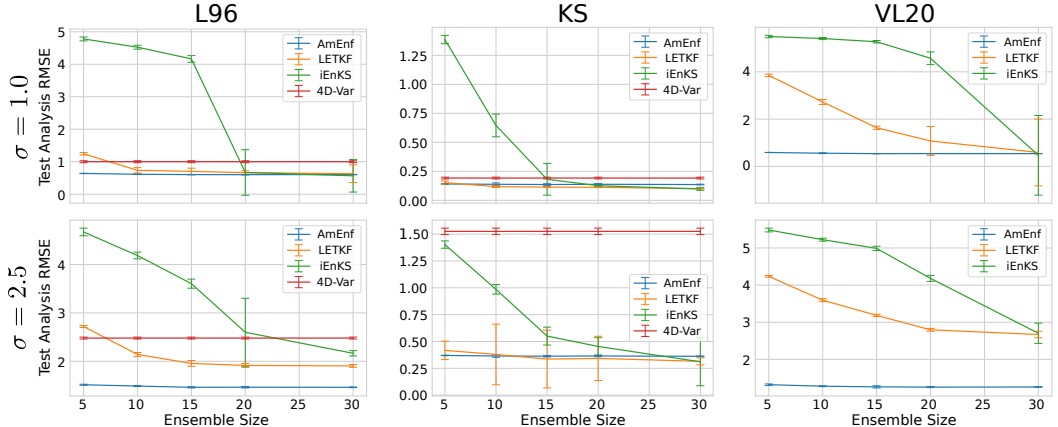

Figure 7: Results from numerical comparisons across multiple ensemble sizes and observation noises.

What is intriguing is that the self-supervised forecast loss actually outperforms the supervised analysis loss for this system. The optimization problem is nonconvex, so neither approach is guaranteed to reach a global minimizer. This is not definitive as the performance gap is within the margin of error; however, it does suggest that there may be some value to including differentiable simulation in the training pipeline beyond simply enabling the denoising property.

## 5.2 Method Comparison

We evaluate performance on three benchmark systems. Loss is reported as the time-averaged RMSE between the mean analysis state estimate and the true state on the test state. Each system is evaluated across a variety of ensemble sizes (denoted by $m$) and isotropic Gaussian observation noise levels (whose standard deviation is denoted by $\sigma$). All results are reported in Figure 7.

We compare performance against a set of widely used filtering methods for data assimilation implemented in the Python DAPPER library [58]. These methods include 4D-Var [24], the Local Ensemble Transform Kalman Filter (LETKF) [59], and the Iterative Ensemble Kalman Smoother (iEnKS) [60]. 4D-Var and the iEnKF are used in a filtering capacity, using only the latest observation rather than a multi-step trajectory. Full experiment settings including hyperparameters such as inflation and localization settings searched are included in the supplementary materials.

**Lorenz 96** For method comparisons, we examine the more challenging case of a partially observed system. At every assimilation step, we observe only a rotating subset consisting of every fourth spatial dimension so that only 25% of the system is observed at any time. These experiments use an RK-4 [61] integrator with step sizes of .05 and partial observations every two integration steps.

**Kuramoto-Shivashinsky** While our theoretical results focus on ordinary differential equations, the method can also be used with partial differential equations like the KS equation. The KS equation [62] is a fourth-order partial differential equation known to exhibit chaotic behavior. In one spatial dimension, the governing equation can be written as:

$$u_t + u_x + u_{xxxx} + uu_x = 0 \tag{11}$$

with periodic boundary conditions. These experiments were fully observed. The system was integrated with an ETD-RK4 method with step sizes of .5 with observations every two steps.

**Vissio-Lucarini 20** The system described by Vissio and Lucarini [63] is a coupled system intended to represent a minimal model of the earth's atmosphere. It augments the Lorenz system with "temperature" variables allowing for more complicated behavior:

$$\begin{aligned}
\frac{dX_j}{dt} &= X_{j-1}(X_{j+1} - X_{j-2}) - \alpha\theta_j - \gamma X_j + F \\
\frac{d\theta_j}{dt} &= X_{j+1}\theta_{j+2} - X_{j-1}\theta_{j-2} + \alpha X_j - \gamma\theta_j + G
\end{aligned} \tag{12}$$

We set $F = 10$, $G = 10$, $\gamma = 1$, $\alpha = 1$, and use 36 spatial points. Experiments using this system are also partially observed with every fourth spatial coordinate being observed at one time. The system is integrated using RK-4 with steps of length .05 and partial observations every two steps. 4D-Var was excluded in this comparison as performance on prior experiments did not justify further examination.

**Results**    As we can see in Figure 7, the AmEnF models match or outperform the competitive methods with more significant improvements coming at smaller ensemble sizes and higher noise levels which are associated with stronger nonlinear effects [60]. The only system in which the AmEnF does not notably outperform the comparison models is the KS equation. This is interesting as our theoretical results rely on assumptions about ODEs while the KS equation is a PDE. Furthermore, the bound we derived has a time component and the KS experiments occur at the largest time scales. Nonetheless, despite these potential theoretical snags, the AmEnF model near matches the the top performing models in accuracy while exhibiting stronger stability under repeated experiments.

## 6    Discussion and Conclusion

**Limitations**    Despite the successful numerical results, the method is not without challenges. The largest consideration is related to scaling. First, while the use of differentiable simulation allowed our method to outperform competitive methods, the availability of adjoints in numerical forecasting is not certain. Forecasts in domains like NWP often use legacy code where adjoints need to be maintained by hand. We expect this to change over time, but this will limit the adoption of similar approaches in the near future. Assuming the presence of adjoints, training cost is also a concern as it requires the repeated simulation of the dynamical systems. However, numerical models tend to change slowly and applications like NWP are among the largest users of compute in the world [1] with assimilation cycles occurring multiple times per day, thus we expect the amortization to pay off over time.

**Conclusion**    We theoretically and experimentally motivate the use of a self-supervised approach for learning to assimilate in chaotic dynamical systems. This approach uses a hybrid deep learning-numerical simulation architecture which outperforms or matches several standard data assimilation methods across several numerical experiments. The performance gains are especially profound at low ensemble sizes, which could enable higher resolution, more accurate simulation for assimilation problems leading to stronger predictions for high value problems like numerical weather prediction in the future.

## Acknowledgments and Disclosure of Funding

This material is based upon work supported by the National Science Foundation under Grant No. 1835825.

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
