# Learning to Assimilate in Chaotic Dynamical Systems: Supplementary Materials

**Michael McCabe**
Department of Computer Science
University of Colorado Boulder
michael.mccabe@colorado.edu

**Jed Brown**
Department of Computer Science
University of Colorado Boulder
jed@jedbrown.org

## A    Notation Table

Table 1: Notation used in paper.

| Symbol | Domain | Description |
|---|---|---|
| $\mathbf{x}(t)$ | $\mathbb{R}^d$ | System state evaluated at time $t$ |
| $g$ | $\mathbb{R}^d \to \mathbb{R}^d$ | $\dot{\mathbf{x}}$ or equivalently the governing dynamics of system |
| $L$ | $\mathbb{R}$ | Lipschitz constant of $g$ |
| $\tau_k$ | $\mathbb{R}$ | Time observation $k$ was recorded |
| $\mathcal{M}_{k:k+1}$ | $\mathbb{R}^d \to \mathbb{R}^d$ | $\mathbf{x}_k + \int_{\tau_k}^{\tau_{k+1}} g(\mathbf{x}(t))\, dt$ |
| $\mathcal{H}_k$ | $\mathbb{R}^d \to \mathbb{R}^o$ | Observation operator used at time index $k$ |
| $\mathbf{y}_k$ | $\mathbb{R}^o$ | Observation value recorded at time index $k$ |
| $\sigma_k^2$ | $\mathbb{R}$ | Variance of isotropic Gaussian observation noise at time index $k$ |
| $\mathbf{x}_k$ | $\mathbb{R}^d$ | $\mathbf{x}(\tau_k)$ |
| $m$ | $\mathbb{Z}$ | Ensemble size |
| $K$ | $\mathbb{Z}$ | Number of observations in assimilation window |
| $\mathbf{x}_{i,k}^f$ | $\mathbb{R}^d$ | Forecast estimate of $\mathbf{x}_k$ from ensemble member $i$ |
| $P_k^f$ | $\mathbb{R}^{d\times d}$ | Forecast estimate of state covariance at time index $k$ |
| $\mathbf{x}_{i,k}^a$ | $\mathbb{R}^d$ | Analysis estimate of $\mathbf{x}_k$ from ensemble member $i$ |
| $P_k^a$ | $\mathbb{R}^{d\times d}$ | Analysis estimate of state covariance at time index $k$ |
| $f_\theta$ | $\mathbb{R}^d \to \mathbb{R}^d$ | Learned analysis update parameterized by weights $\theta \in \mathbb{R}^p$ |
| $\mathcal{L}^a$ | $\mathbb{R}^d \to \mathbb{R}$ | Supervised analysis loss |
| $\mathcal{L}^f$ | $\mathbb{R}^d \to \mathbb{R}$ | Supervised forecast loss |
| $\mathcal{L}^{ssf}$ | $\mathbb{R}^o \to \mathbb{R}$ | Self-supervised forecast loss |
| $\mathcal{L}^{ssa}$ | $\mathbb{R}^o \to \mathbb{R}$ | Self-supervised analysis loss |

## B    Connections to Classical Assimilation Methods

The Amortized Ensemble Filters (AmEnF) used in our experiments have strong connections to both ensemble-based sequential filtering methods and variational assimilation. We discuss both here.

### B.1    Connections to the EnKF

As we mentioned in the main text of the paper, the AmEnF architecture can be seen as directly replacing the analysis update equations from the EnKF with an optimized neural network. There are a large number of EnKF variants, but the most straightforward to describe is the stochastic EnKF, which "samples" a unique observation corresponding to each ensemble member from the observation noise

35th Conference on Neural Information Processing Systems (NeurIPS 2021).

distribution. Letting $H_k$ be the linearization of our observation operator $\mathcal{H}_k$, the update equations for the stochastic EnKF are:

$$
\begin{aligned}
\textbf{Forecast:} \quad & \mathbf{x}_{i,k}^f = \mathcal{M}_{k-1:k}(\mathbf{x}_{i,k}^f), & i = 1, 2, \ldots, m \\
\textbf{Analysis:} \quad & P_k^f = Cov(\{\mathbf{x}_{i,k}^f\}_{i=1}^m) \\
& \tilde{\mathbf{y}}_{i,k} = \mathbf{y} + \boldsymbol{\epsilon}_i, & \boldsymbol{\epsilon}_i \sim \mathcal{N}(0, \sigma_k^2 I) \\
& K_k = P_k^f H_k^T (H_k P_k^f H_k^T + \sigma_k^2 I)^{-1} \\
& \mathbf{x}_{i,k}^a = \mathbf{x}_{i,k}^f + K_k(\tilde{\mathbf{y}}_{i,k} - H_k \mathbf{x}_{i,k}^f).
\end{aligned}
\tag{1}
$$

We emphasize that this formulation does not lead to an efficient implementation and use it only for explanatory purposes.

As with most ensemble-based filters, the forecast step is identical between the AmEnF and the EnKF. The difference comes from the analysis step. The EnKF update is the Kalman filter update with the Kalman gain $K$ computed using the empirical covariance of the ensemble. The Kalman filter formula can be derived both from Bayesian statistics and through a variational principle as the optimal update in the case that both the observation and state distributions are Gaussian [1].

As the image of a Gaussian random variable under nonlinear transformation is not guaranteed to be Gaussian, this update is no longer optimal under all nonlinear dynamics. Amortized assimilation replaces this pre-specified update with a learned update which we optimize using historical data. The amortized model uses the same basic inputs (the ensemble members, the observation, and the covariance) with a small number of additional features included due to the ease of doing so.

## B.2 Connections to Weak-Constraint 4D-Var

Weak-constraint 4D-Var [2] is a variation of classical, or strong-constraint, 4D-Var which accounts for model inaccuracy by performing optimization over the full state trajectory $\mathbf{x}_{0:K}$ within an assimilation window rather than just finding initial conditions $\mathbf{x}_0$ and using the time evolution of those initial conditions for all likelihood calculations. This results in the following objective function:

$$
\begin{aligned}
\mathcal{J}(\mathbf{x}_{0:K}) = & (\mathbf{x}_0 - \mathbf{x}_b)^T B^{-1}(\mathbf{x}_0 - \mathbf{x}_b) \\
& + \sum_{k=1}^K (\mathcal{H}_k(\mathbf{x}_k) - \mathbf{y}_k)^T R_k^{-1}(\mathcal{H}_k(\mathbf{x}_k) - \mathbf{y}_k) \\
& + \sum_{k=0}^{K-1} (\mathbf{x}_{k+1} - \mathcal{M}_{k:k+1}(\mathbf{x}_k))^T Q_{k+1}^{-1}(\mathbf{x}_{k+1} - \mathcal{M}_{k:k+1}(\mathbf{x}_k))
\end{aligned}
\tag{2}
$$

where the subscript $b$ indicates a prior or background estimate and the matrices $B, R_k, Q_k$ indicate the background, observation, and model error covariances respectively. On the surface, this looks quite different from the objective used in our work. However, it actually only differs by the inclusion of a prior term prior to unrolling the self-supervised amortized assimilation objective. If we included a prior based on the ensemble distribution at each step of the amortized assimilation objective (which was done in an earlier iteration of our approach, though we typically observed worse performance compared to the presented model) and assume $R_k = I$, the objective could be re-written as:

$$
\begin{aligned}
\mathcal{J}(\mathbf{x}_{0:K}^a) = & \sum_{k=1}^K (\mathcal{H}_k(\mathbf{x}_k^a) - \mathbf{y}_k)^T R_k^{-1}(\mathcal{H}_k(\mathbf{x}_k^a) - \mathbf{y}_k) \\
& + \sum_{k=0}^{K-1} (\mathbf{x}_{k+1}^a - \mathcal{M}_{k:k+1}(\mathbf{x}_k))^T {P_{k+1}^f}^{-1}(\mathbf{x}_{k+1} - \mathcal{M}_{k:k+1}(\mathbf{x}_k))
\end{aligned}
\tag{3}
$$

with all $\mathbf{x}_k$ generated through the sequential filtering process. This objective differs from the weak-constraint 4D-Var objective with $Q_k = P_k^f$ only by the initial prior. $Q_k = P_k^f$ is not an especially sensible choice for model error covariance and the mechanics of generating the predicted trajectories are quite different (direct optimization vs. sequential updates), but the connection is nonetheless interesting as it indicates that our method may be interpreted as a direct amortization of the weak-constraint 4D-Var procedure under certain conditions.

## C  Proofs

### C.1  Lemma 1

**Lemma 1** (*Noise2Self – Restricted*). *Suppose concatenated noisy observation vector $\mathbf{y} = [\mathbf{y}_k; \mathbf{y}_{k+1}]$ is an unbiased estimator of concatenated state vector $\mathbf{x} = [\mathbf{x}_k; \mathbf{x}_{k+1}]$ and that the noise in $\mathbf{y}_k$ is independent from the noise in $\mathbf{y}_{k+1}$. Now let $\mathbf{z} = f(\mathbf{y}) = [\mathbf{z}_k; \mathbf{z}_{k+1}]$. If $f$ is a function such that $z_{k+1}$ does not depend on the value of $\mathbf{y}_{k+1}$ then:*

$$\mathbb{E}_{\mathbf{y}}\|f(\mathbf{y})_{k+1} - \mathbf{y}_{k+1}\|^2 = \mathbb{E}_{\mathbf{y}}[\|f(\mathbf{y})_{k+1} - \mathbf{x}_{k+1}\|^2 + \|\mathbf{y}_{k+1} - \mathbf{x}_{k+1}\|^2] \tag{4}$$

*Proof:*

$$\mathbb{E}_{\mathbf{y}}\|f(\mathbf{y})_{k+1} - \mathbf{y}_{k+1}\|^2 = \mathbb{E}_{\mathbf{y}}\|(f(\mathbf{y})_{k+1} - \mathbf{x}_{k+1}) - (\mathbf{y}_{k+1} - \mathbf{x}_{k+1})\|^2] \tag{5}$$

$$= \mathbb{E}_{\mathbf{y}}[\|f(\mathbf{y})_{k+1} - \mathbf{x}_{k+1}\|^2 + \|\mathbf{y}_{k+1} - \mathbf{x}_{k+1}\|^2 - 2\langle f(\mathbf{y})_{k+1} - \mathbf{x}_{k+1}, \mathbf{y}_{k+1} - \mathbf{x}_{k+1}\rangle] \tag{6}$$

$$= \mathbb{E}_{\mathbf{y}}[\|f(\mathbf{y})_{k+1} - \mathbf{x}_{k+1}\|^2 + \|\mathbf{y}_{k+1} - \mathbf{x}_{k+1}\|^2] - 2\sum_{j=1}^{d}\mathbb{E}_{\mathbf{y}}[(f(\mathbf{y})_{k+1}^{(j)} - \mathbf{x}_{k+1}^{(j)})(\mathbf{y}_{k+1}^{(j)} - \mathbf{x}_{k+1}^{(j)})] \tag{7}$$

$$= \mathbb{E}_{\mathbf{y}}[\|f(\mathbf{y})_{k+1} - \mathbf{x}_{k+1}\|^2 + \|\mathbf{y}_{k+1} - \mathbf{x}_{k+1}\|^2] - 2\sum_{j=1}^{d}\mathbb{E}_{\mathbf{y}}[f(\mathbf{y})_{k+1}^{(j)} - \mathbf{x}_{k+1}^{(j)}]\mathbb{E}_{\mathbf{y}}[\mathbf{y}_{k+1}^{(j)} - \mathbf{x}_{k+1}^{(j)}] \tag{8}$$

$$= \mathbb{E}_{\mathbf{y}}[\|f(\mathbf{y}_{k+1}) - \mathbf{x}_{k+1}\|^2 + \|\mathbf{y}_{k+1} - \mathbf{x}_{k+1}\|^2]. \tag{9}$$

Note that the superscript $j$ indicates vector element indices while the subscript corresponds to block entries corresponding to time indices. Statement (7) uses Fubini's theorem to move the expectation integral into the summation. We then use the assumed independence of $f(y)_{k+1}^{(j)}$ and $y_{k+1}^{(j)}$ to convert the expectation of the product of independent random variables into the product of expectations. Finally, $\mathbb{E}_{\mathbf{y}}[\mathbf{y}_{k+1}^{(j)} - \mathbf{x}_{k+1}^{(j)}] = 0$ as $\mathbf{y}_{k+1}^{(j)}$ is an unbiased estimator of $\mathbf{x}_{k+1}^{(j)}$, resulting in the final term vanishing and giving us the stated identity.

### C.2  Proposition 1

**Proposition 1** (Zero Loss Regime). *Under the stated assumptions, let $f_\theta$ denote a family of functions parameterized by $\theta$ for which $\min_\theta \mathcal{L}^f(\theta) = 0$. For any $\theta$ which achieves this minimum, it is also true that $\mathcal{L}^a(\theta) = 0$ and that $\theta$ is in set of minimizers of $\mathbb{E}_{\mathbf{y}_{0:K}}[\mathcal{L}^{ssf}(\theta)]$.*

*Proof:* Plug in $\mathcal{M}_{k:k+1} \circ f_\theta(\mathbf{x}_k^f, \mathbf{y}_k)$ as the function used in Lemma 1. This obeys the assumptions that $f(\mathbf{y}_k)$ is independent from $\mathbf{y}_{k+1}$ as observation noise is assumed to be uncorrelated in time and the composition predicts $\mathbf{y}_{k+1}$ based entirely on prior information. Since the supervised component of the loss is zero, the expected self-supervised loss consists only of the irreducible variance term and thus is also minimized.

For the analysis loss, assumption 1 gives us that the solution to the initial value problem is unique and therefore the preimage of $\mathcal{M}_{k:k+1}$ contains a single point for all inputs in the range of $\mathcal{M}_{k:k+1}$. Since our forecast is produced by applying the operator, the forecast is clearly in the range of the operator. Thus, if $\mathcal{M}_{k:k+1} \circ f_\theta(\cdot) = x_{k+1}$ then $f_\theta(\cdot) = x_k$ resulting in an analysis loss of zero as well.

### C.3  Proposition 2

**Proposition 2** (Non-zero Loss Regime). *Under the previously stated assumptions, the supervised analysis loss can be bounded by the supervised forecast loss as:*

$$\frac{1}{K-1}\sum_{k=1}^{K-1}\left\|\left(\frac{1}{m}\sum_{i=1}^{m}f_\theta(\cdot)\right) - \mathbf{x}_k\right\| \leq \frac{e^{L(\tau_{k+1}-\tau_k)}}{K-1}\sum_{k=1}^{K-1}\max_{i\in[m]}\left\|\mathcal{M}_{k:k+1}\circ f_\theta(\mathbf{x}_{i,k}^f, \cdot) - \mathbf{x}_k\right\| \tag{10}$$

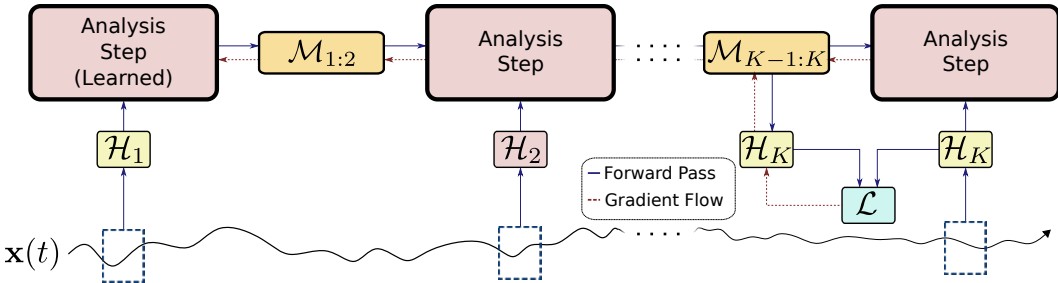

Figure 2: The process for learning to use unknown observation operators. In this example $\mathcal{H}_2$ is learned from data.

*Proof:* We construct the bound using the Lypanuov spectrum of the system defined by dynamics $g$. The Lyapunov exponents of the system characterize the rate of separation of perturbed system states. Since we are bounding the separation at an earlier time point by the separation at a later time, the relevant quantity is the the minimal Lyapunov exponent $\lambda_1$, which assuming the value is negative, bounds the rate at which perturbations in the initial conditions can shrink:

$$\left\| \mathcal{M}_{k:k+1} \circ f_\theta(\mathbf{x}_{i,k}^f, \cdot) - \mathbf{x}_{k+1} \right\| \geq e^{\lambda_1(\tau_{k+1} - \tau_k)} \left\| f_\theta(\mathbf{x}_{i,k}^f, \cdot) - \mathbf{x}_k \right\| \tag{11}$$

The Lyapunov exponents are inherently local properties, but as $g$ is $L$-Lipschitz, we can globally bound the Lyapnuov exponents $|\lambda_i| \leq L$ for all $i = 1, \ldots, d$. Plugging this in and rearranging, we get:

$$\left\| f_\theta(\mathbf{x}_{i,k}^f, \cdot) - \mathbf{x}_k \right\| \leq e^{L(\tau_{k+1} - \tau_k)} \left\| \mathcal{M}_{k:k+1} \circ f_\theta(\mathbf{x}_{i,k}^f, \cdot) - \mathbf{x}_{k+1} \right\| \tag{12}$$

which using linearity and the fact that the maximum distance from an ensemble member cannot exceed the distance from the mean of the ensemble yields the stated bound.

# D Incomplete or Unknown Observations

## D.1 Incomplete Observations

We experimented with two approaches for incorporating partial observations with known observation operators into convolutional architectures. In both, we incorporated an additional channel with indicators of .1 if the coordinate was observed and -.1 if the coordinate was not observed. The simpler of the two approaches was simply to pass a value of zero in the observation channel for all unobserved coordinate.

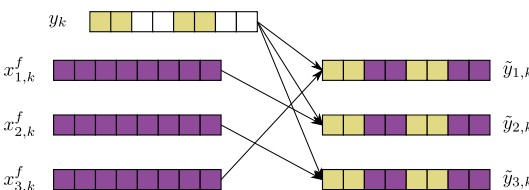

Figure 1: Partial observations were included in the convolutional architecture by replacing missing values with forecast values from a shuffled ensemble.

The second approach, which was used in our experiments, used partial observations as an opportunity to cross-pollinate the update with information from other ensemble members. In this approach (Figure 1), each observation channel consisted of the actual observed values in the coordinates that were observed while the missing information was replaced with the forecast values from other ensemble members. The information from the ensemble was treated as though it was part of the observation in that the imputed values were detached from the computation graph so that no gradient information flowed backwards through this process.

## D.2 Unknown Observation Operators

While not the focus of our work, amortized assimilation does admit a natural approach for learning observation operators which we describe here (Figure 2). We do not evaluate the loss based on

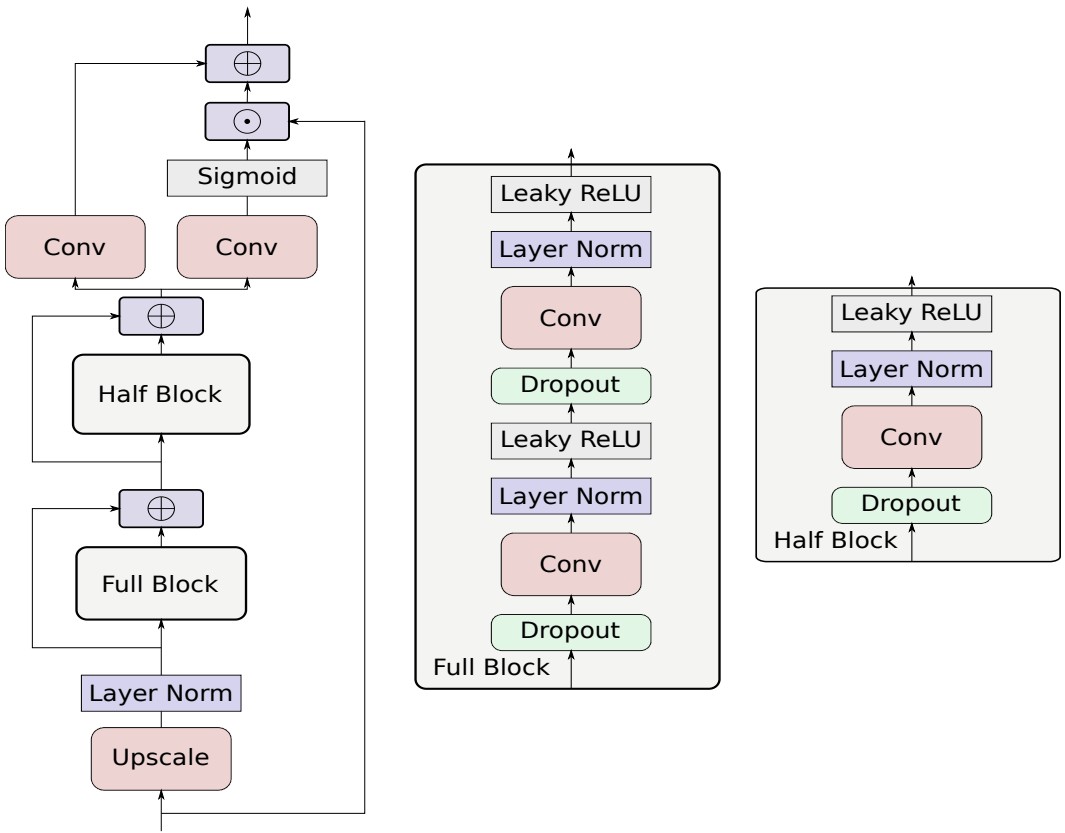

Figure 3: Outline of neural network architecture used for learned assimilation step in all experiments.

unknown observation operators. Instead, the unknown observation type is processed by an additional independent neural network whose architecture may depend on the data type of the unknown input. This learned observation operator is then trained using only gradient information flowing backwards through the differentiable simulation from losses evaluated using future, known observation types. This enables direct assimilation of data types without direct mapping to state variables like hyperspectral imaging or other satellite-derived information.

# E   Experiment Settings

## E.1   L96 and VL20

All experiments used the same architecture outline as Figure 3. For L96 and VL20, all convolutional filters had width 5 with circular padding of 2. Each layer, apart from the final readout layer, used 64 convolutional filters. Six channels of memory were used. Dropout was set to .2. We used a covariance width of 3, indicating that channels were included for the coordinate variance and covariances with the coordinates to the left and right of the current entry. Training was done using mini-batches of size 64.

## E.2   KS Equation

The KS experiments used the same architecture as the L96 and VL20 experiments but with slightly different hyperparameter settings. For these experiments, we used a 64 filters per layer with a filter size of 7 and circular padding of size 3. The dropout rate was set to .2 and the covariance width was set to 5. Memory depth was set to 4. The system was discretized uniformly into 128 spatial coordinates and training used mini-batches of size 32.

### E.3 Hyperparameter Search

We tuned the comparison methods on validation sets generated from 5 different initial conditions through a grid search. This grid search was performed independently for each tested dynamical system, ensemble size, and noise level. For testing, the parameters giving the highest average RMSE over the starting conditions were used. The parameters searched are listed in table 2.

The AmEnF was tuned in a more ad-hoc manner with variables moved up or down the ladder of available options when instabilities occurred. This primarily affected the initial learning rate as we found the approach was largely agnostic to sensible settings of the dropout rate while performance generally improved with longer sequences during training at the expense of additional memory consumption. Note: for the AmEnF, this only includes parameters explored while tuning the presented architecture. Decisions on number of filters and batch sizes were made to fully utilize available GPU memory subject to the state size of the dynamical system, the number of unrolled steps, and the ensemble size. All AmEnF models were trained with an ensemble size of $m = 10$ and tested using ensembles of various sizes.

Table 2: Hyperparameter search space.

| Method | Parameter | Range |
|--------|-----------|-------|
| 4D-Var | B | {Historical Covariance, $I$} |
|  | B-Scale | $\{.1, .2, .4, 1, 1.25, 1.5, .75, 2, 2.25, 2.5, 3\}$ |
| LETKF | Inflation | $\{1.00, 1.01, 1.02, 1.04, 1.07, 1.1, 1.2, 1.4, 1.7, 2.00\}$ |
|  | Rotation | {True, False} |
|  | Localization Radius | $\{0.1, 0.2, 0.4, 0.7, 1.0, 2.0, 4.0, 7.0, 10.0, 20.0, 40.0, 70.0\}$ |
| iEnKS | Inflation | $\{1.00, 1.01, 1.02, 1.04, 1.07, 1.1, 1.2, 1.4, 1.7, 2.00\}$ |
|  | Rotation | {True, False} |
| AmEnF | Dropout Rate | $\{.1, .2, .3, .5\}$ |
|  | Initial Learning Rate | $\{1e-2, 5e-3, 1e-3, 8e-4, 5e-4, 3e-4, 1e-4\}$ |
|  | Steps per Training Seq | $\{1, 10, 20, 40\}$ |