# OpenReview forum: "Learning to Assimilate in Chaotic Dynamical Systems"
_NeurIPS.cc/2021/Conference — NeurIPS 2021 Poster_

### Official Review · Reviewer_UkND · 2021-07-12

**Rating:** 6
**Confidence:** 3

**Summary:**

The authors introduce amortized assimilation or AmEnF, a novel method for inferring dynamics from sampled states, and predicting initial and future states from it.

**Limitations And Societal Impact:**

I see no negative societal impact from this work.

**Main Review:**

The key idea of AmEnF is that we can run the model from several different timepoint and compare the different predictions we get from the model to one another, results in a self-supervised loss. They prove that the self supervised loss is bounded by the standard supervised loss. I think this is a really neat idea, and an interesting new direction for self-supervised learning for dynamical system.

The authors show that their model performs well on standard data assimilation tasks such as Lorenz 96, Vissio-Lucarini 20 and Kuramoto-Shivashinsk, especially when observed data is sparse and noise is high, which are usually the instances where self-supervised are especially helpful.

Overall, the paper does a good job at explaining the background, previous works, and their innovation and contributions.

Minor remarks: All figures are to small and a quite hard to read, especially figure Figure 1.

I recommend acceptance.

**Time Spent Reviewing:**

2 hours

---

> ### Author Response · Authors · 2021-08-06
> **Re: Official Review of Paper1020 by Reviewer UkND**
>
> Thanks for the review! We completely agree on some of the figures - some of them rendered smaller than we expected and we didn't have the time pre-submission to fix it. There's a few that should be trivial to expand (ie, no restructuring elsewhere needed) due to large amounts of white space. Others (figure 1, for instance) will be a bit more complicated, but it seems doable.
>
> If there are any concerns you have that we might be able to address that would make you comfortable increasing your rating of the paper, please let us know and we'll do our best to address them. Otherwise, we appreciate your effort as a reviewer and we're happy to see that you enjoyed our paper.

---

### Official Review · Reviewer_kG8H · 2021-07-15

**Rating:** 6
**Confidence:** 5

**Summary:**

The paper presents, amortized assimilation, way to perform data assimilation through deep learning utilizing self-supervised training to obtain an unbiased estimate of the true state.

**Limitations And Societal Impact:**

Yes

**Main Review:**

The abstract of the paper is overwhelmed with technical jargons. While the need for technical jargons maybe be essential at most points in a paper, I would expect the authors to try and write a more easy-to-understand abstract so that the work may appeal to both the deep learning and the DA community.

"Line 44" what is oft-repeated in numerical simulations of Earth system ? I do not think that there are any oft-repeated processes, else the authors should cite something that explains this.

Lines 51-60 are extremely hard to read. In fact, throughout the paper, the authors might want to keep in mind that their application is useful for both ML and DA audiences and can rephrase the end of their introduction to read a little more easily to a vaster range of audience, say from geoscience, DA, dynamical systems, and physics along with ML.

"Lines 102-106" At small ensemble sizes DA has been successfully used for even the most complex NWPs through localization and inflation.  Generating large ensembles has also been successfully performed with deep learning methods as well. The statements in between these lines are not enough to showcase a need for the authors' method. A clear motivation is either  absent in this introduction or might get hidden owing to the overly technical writing .

"Lines 112-113" Sequential filters do not respect conservation laws ? When making claims like these, the authors should cite papers or show evidence.

"that we view sequential filtering as a particular recurrent prediction model which can be unrolled into a multi-step computation graph allowing us the train using backpropogation through time" -- What is this computation graph? While the authors present a very well thought out method, with enough novelty, the writing makes it very hard to follow. This is not helping the authors make their case for the importance or relevance of this method

The results are interesting but does not show the advantage of this method over other traditional DA methods. What happens with small ensemble sizes and localization and inflation? In the absence of fair comparisons , the results are not complete.

References: No journal names in the references

[39] Julien Brajard, Alberto Carassi, Marc Bocquet, and Laurent Bertino. Combining data assimilation and machine learning to emulate a dynamical model from sparse and noisy observations: a case study with the lorenz 96 model, 2020. -- incomplete reference, where's the journal name? https://gmd.copernicus.org/preprints/gmd-2019-136/

 [42] Ashesh Chattopadhyay, Mustafa Mustafa, Pedram Hassanzadeh, Eviatar Bach, and Karthik
 Kashinath. Towards physically consistent data-driven weather forecasting: Integrating data  assimilation with equivariance-preserving deep spatial transformers, 2021. --- same problem. Here's the paper: https://gmd.copernicus.org/preprints/gmd-2021-71/

With an improvement in the writing, and more experiments with comparisons with DA+localization and inflation, this will be a very good contribution.







**Time Spent Reviewing:**

4

---

> ### Author Response · Authors · 2021-08-05
> **Re: official review**
>
> ## General Response
>
> First off, thank you for the extremely detailed and constructive feedback. Your expertise on the subject is clear and future drafts of the paper will certainly benefit from your feedback.
>
> As this is a rebuttal and the goal is ultimately to try to convince you to improve your score, there is one item we’d like to address before going into line-by-line responses and that is accessibility. Accessibility is something we wrestled with in writing the paper and isn’t necessarily something that we’ll be able to address completely within the constraints of the publishing format, but we hope that by explaining our thought process on some of these points, you might be willing to ease some of your judgement on this particular issue.
>
> Fundamentally, the paper straddles two communities - the problem itself comes from the DA community, but the methodology and analysis are derived from a line of research (self-supervised denoising) that grew out of deep learning (computer vision in particular). The work in this paper is largely focused on developing the specific deep learning approach and providing analysis of that approach and so we chose to submit to deep learning venues rather than data assimilation ones.
>
> Given space limitations and given that this is a deep learning venue, we did strongly lean towards language that is more common in the deep learning community. Responses to an earlier draft made it clear that how our work fits into the broader deep learning ecosystem was a major concern to DL oriented reviewers and so we made an effort to ensure our submission was firmly rooted in prior deep learning research. Unfortunately, that means that a large amount of explanatory space was spent describing the basics of the DA problem while some deep learning language is used with only cursory definitions. However, there are some definitions for jargon that were clearly missed in the truncation and we point those out in our line-by-line responses.
>
> ## Line by Line
>
> There's one point we want to bring to the top since this is a misunderstanding which we can address by adding a few words and has a large impact on perceived integrity of the experiments:
>
> > The results are interesting but does not show the advantage of this method over other traditional DA methods. What happens with small ensemble sizes and localization and inflation? In the absence of fair comparisons , the results are not complete.
>
> The reference models in this case are actually using localization and inflation. This unfortunately is not explicitly mentioned in the main body of the submitted version (though it has been fixed), but the hyperparameters mentioned in line 311 are actually mostly referring to the DA methods where a large grid search was performed over inflation/localization settings at each ensemble size which is documented in the appendix.
>
> Now onto other concerns:
>
> >The abstract of the paper is overwhelmed with technical jargons. While the need for technical jargons maybe be essential at most points in a paper, I would expect the authors to try and write a more easy-to-understand abstract so that the work may appeal to both the deep learning and the DA community.
>
> The terms that we feel are relatively jargony are:
> - Self-supervised - this can likely be addressed by including the definition immediately rather than later as a rephrasing and not directly connecting it (in this case we’re using it to refer to learning from the noisy data generated by the system rather than ground truth).
> -Differentiable simulation - This term is fairly prominent in DL for computational science, but given that it’s used in conjunction with other field-specific terms, the first usage could be replaced with its actual description.
> - Gradient flow - The term information flow would be less technically accurate, but we think it might do a better job of conveying the intent to a general reader.
> - Recurrent memory - this one is fairly hard to replace as it’s using prior knowledge of LSTM/GRU type architectures to condense the explanation. We could refer to it as an augmented state, but we suspect that would cause more confusion for DL readers. This one will likely need to stay as is though we could specifically reference LSTM-style memory to make it clearer what we're referring to.
>
> >"Line 44" what is oft-repeated in numerical simulations of Earth system ? I do not think that there are any oft-repeated processes, else the authors should cite something that explains this.
>
> The oft-repeated process in this case is variational data assimilation. The parallel being drawn here is between methods like amortized inference which replaces the variational parameter fitting process of traditional variational inference methods with a neural network that maps from inputs to distribution parameters. In our case, rather than solving the nonlinear 4DVar optimization problem over every assimilation window, our amortized model learned to directly map from inputs (forecast state and observations plus the additional inputs) to our analysis estimate.
>
> >Lines 51-60 are extremely hard to read. In fact, throughout the paper, the authors might want to keep in mind that their application is useful for both ML and DA audiences and can rephrase the end of their introduction to read a little more easily to a vaster range of audience, say from geoscience, DA, dynamical systems, and physics along with ML.
>
> For 55-60, this was intended to be more of a list of items that we’ll go into more detail on in the analysis, but from a clarity perspective, it would likely benefit from a higher level view with a specific reference to where the topic is discussed as there isn’t enough space in the introduction to provide enough background for the low-level list to make sense.
>
> For 51-55, as none of those assumptions have actually been introduced at that section of the paper, the statement comes across as extremely vague. The goal was to say essentially “the assumptions we make to fit DA into the self-supervised denoising framework are reasonable and fairly standard in typical DA analysis” but that level of detail isn’t necessary or useful at this point in the paper. As on re-read, it’s clear that we haven’t actually defined what we mean by self-supervised denoising at this stage, we plan on revising this paragraph to focus on what self-supervised learning is and how we use differentiable simulation to accomplish it.
>
> >"Lines 102-106" At small ensemble sizes DA has been successfully used for even the most complex NWPs through localization and inflation. Generating large ensembles has also been successfully performed with deep learning methods as well. The statements in between these lines are not enough to showcase a need for the authors' method. A clear motivation is either absent in this introduction or might get hidden owing to the overly technical writing .
>
> Our goal in this section is less to say that assimilation cannot be done at small ensemble sizes (given that as you pointed out - it clearly has on real world systems) and more that our goal is to improve results in that setting. This is more targeted at ML-readers, particularly those coming from fields where particle filtering methods are more widespread, as an explanation as to why performance at smaller ensemble sizes is significant. We would note that this is the domain where our experiments did outperform the reference methods (which did incorporate localization and inflation as we explain in the later comment).
>
> >"Lines 112-113" Sequential filters do not respect conservation laws ? When making claims like these, the authors should cite papers or show evidence.
> That’s a good point. There used to be a larger paragraph comparing variational and EnKF-type methods in the introduction where this was highlighted (and cited) as an advantage of variational methods over the EnKF so at that point, 112-113 was a callback to that earlier statement, but without that larger exposition block which was removed due to space constraints, a citation is certainly necessary here.
>
> >"that we view sequential filtering as a particular recurrent prediction model which can be unrolled into a multi-step computation graph allowing us the train using backpropogation through time" -- What is this computation graph? While the authors present a very well thought out method, with enough novelty, the writing makes it very hard to follow. This is not helping the authors make their case for the importance or relevance of this method
>
> This should reference Figure 4 which is an example of the graph in question. This explanation is there to set up the contrast with Haarnoja et al. which is a related earlier paper that parameterized certain components of a vanilla Kalman filter with neural networks so in essence we’re saying “this is what is shared, but…”. This could likely be broken out into two sentences where the first describes the standard RNN truncated compute graph perspective and the second describes the contract with Haarnoja et al.
>
>
> >References: No journal names in the references
>
> Thanks for this. This is all automatically generated bibtex, so we’re a little confused how that happened, but we’ll have to iterate to make sure the other entries are also accurate.

---

> > ### Comment · Reviewer_kG8H · 2021-08-05
> > **The authors have given compelling responses . Changed my rating**
> >
> > The authors do a great job with their rebuttal. My main issue was really about a fair comparison with traditional DA+localization+inflation.
> >
> > 1. Regarding localization: Based on the authors' rebuttal, I went through the supplemental. Yes, there is a mention of localization radius. I understand that all their experiments were with localization. Now, with ensemble size = 5, it seems that localization actually does a great job (the difference in error for LETKF in comparison to the proposed method is quite small). Now, I'm unsure of what that value of RMSE means (Maybe the authors have explained that somewhere in the paper, I did have trouble following the flow of material... anyway). Does this difference in error really reflect anything on the free prediction performance? -- Note, that the whole point of DA is to get an analysis state that can be used as an initial condition for free prediction, so how much does that difference in error affect the prediction horizon.
> >
> > 2. The whole EnKF does not respect mass conservation: I didn't understand the rebuttal. Do the authors mean that they have a paper that shows that ? or there is some evidence somewhere? Anyway, it seemed interesting so I was able to dig up a paper. Maybe this should be put in the manuscript, or not, whatever the authors feel. https://journals.ametsoc.org/view/journals/mwre/131/11/1520-0493_2003_131_2595_tmocpl_2.0.co_2.xml
> >
> > 3. The case about particle filters is quite compelling. I agree with the authors that for such DA algorithms, localization is too ad-hoc and deep learning-based methods can definitely benefit that community. Please discuss that if you haven't already done so.
> >
> > Anyway, based on the rebuttal I'm quite sold on the fact that the experiments are quite interesting and despite (a lot of ) concerns about how much the method would be useful for real systems (beyond L96 and KS) and keeping in mind physicists in this domain have only faced issues with scaling DA algorithms (especially particle filters) for high-dimensional GFD applications, this is a novel and interesting method and truly contributes to the ML+DA community which is growing every day and solving important and relevant problems. This paper is worth being discussed and would lead to further developments in this community. I thank the authors for the detailed rebuttal.

---

> > > ### Author Response · Authors · 2021-08-06
> > > **Thanks!**
> > >
> > > Thank you for the revision! We're happy to answer any additional questions. We'll use the same numbering format from your questions.
> > >
> > > 1. I think this is true on the KS experiments (where the LETKF actually outperformed our method, though by a neglible margin), but on the partially observed L96 and VL20 experiments, the gap was still fairly large at ensemble size of 5. Admittedly VL20 is an extension of L96 so one might expect them to have similar results and cannot entirely discount KS as an outlier in that regard. Fortunately (for our method at least), the results are still comparable (and seemingly more stable between noise samples) to the LETKF results. The RMSE reported here is the root mean squared error between the true state and the analysis state estimate averaged over time (and over coordinates so the results are directly comparable to observation noise) which we could definitely make more explicit.
> > > 2. Thanks. We've updated our paper to include the citation since we agree that the claim should be supported. Our statement was originally referring to an article comparing EnKF and 4DVar methods where the ability to incorporate conservation terms was mentioned offhandedly as an advantage for 4DVar. This doesn't appear to be the same article, but it makes the same argument: https://www.ecmwf.int/sites/default/files/elibrary/2003/10817-relative-merits-4d-var-and-ensemble-kalman-filter.pdf (section 4, table 1).
> > >
> > > The concern over scaling is certainly fair and addressing it is something we see as a major direction for future work. This paper was more focused on deriving the connection between self-supervised denoising and DA and demonstrating that neural networks built on that connection can lead to strong DA results, but as with most tasks, there's a ways to go between proof-of-concept and full scale operationalization.

---

### Official Review · Reviewer_NmPU · 2021-07-19

**Rating:** 6
**Confidence:** 3

**Summary:**

The overarching goal of this paper is to improve assimilation of (chaotic) dynamical systems. The authors introduced a self-supervised framework, amortized assimilation, which uses methods from Ensemble filters and supervised denoising.
 In Section 3, they develop their own method, first focusing on amortized ensemble filters and then deriving their self-supervised assimilation method. In Section 4, their method is compared with standard data assimilation methods on chaotic benchmarks (such as the Lorenz 96).

**Limitations And Societal Impact:**

Yes

**Main Review:**

1) I cannot really comment on the Originality of this paper. The methods they use seem common in the dynamical system literature (which they do not review), but maybe there is a crucial difference when moving to assimilation I overlook. Thus I don’t feel confident enough to make a statement.

2) I found the paper to be hard to understand. Some points are definitely lack of knowledge on my part, but other parts could probably be better explained

- The state that the problem they want to solve is a high-quality estimate of the initial state (see abstract). They actually never explain how they estimate the initial state with their model, why their model should be better suited to estimate the initial value nor do they compare a measure of quality for the initial state in their empirical section.

- In the abstract they claim that they use "independent neural networks to assimilate specific observation types”(l. 8f), but later they claim that they didn’t do that, but instead handled them as follows:
(Line 226 )“Coordinates which are not observed can be masked and an additional channel is appending indicating whether a particular coordinate was observed during the given assimilation cycle. This is depicted by the shading in the observation channels of Figure 3. More details on the masking process can be found in the supplementary material"

Either I misunderstand the authors, or their abstract and later explanations do not match …

- They claim:”
(Line 251)  All methods tested have explicit knowledge of the system dynamics. Due to the challenge of learning chaotic dynamics, we found that methods which rely on learning the dynamics were uncompetitive”
I am not sure that is true ...

- Their core section about self-supervised assimilation could be a lot clearer. At one point the authors write "letting us use the framework from [16] to analyze the problem” without further explaining the framework …
This makes it very arduous for the reader to understand what they do

- Figure 6 should have a legend or a more elaborate caption. The plot is not understandable without reading the text and even in the text the colors of the lines plotted is not explained …

3) I should point out some comments that need to be addressed (please see below):

- First, there are some grammatical mistakes as well as missing words and typos …

- Figure labels especially in Figure 7 should be a lot larger. I had to zoom a lot to see the figures

- They should explain what they mean by “independent neural networks to assimilate specific observation types” in the abstract and how that compares to line 226ff (see above)

- If possible they should further explain, why they focus so much on the error in the estimate of the initial state of the system (abstract & introduction) as opposed to other model error and why they do not compare the quality of initial state estimation in their experimental section, if that estimation is so essential

- Lastly, in their subsection Training objective, the authors should consider more training-runs. They find an “intriguing” result but because they only consider one training-run (why???) it’s entirely unclear whether this is to chance or an significant effect.

**Time Spent Reviewing:**

5 hours

---

> ### Author Response · Authors · 2021-08-06
> **Re: Official Review of Paper1020 by Reviewer NmPU**
>
> ## General
>
> Thanks so much for the valuable review. You’ve pointed out several issues that we missed in revisions that we can strengthen the paper by tackling. Our goal for this rebuttal is to address some of your questions, tell you our plans for correcting the more mechanical issues (plots, wording), and generally convince you that we can address enough of your concerns within the current submission that you might consider increasing your rating.
>
> We feel a number of your comments relate to similar issues, so rather than going strictly line-by-line, we will group them by topic.
>
> ## Initial condition estimation
>
> Lines 27-30 are where the importance of initial condition estimation is discussed, but it seems there is some confusion over precisely what we mean here. Somewhat counterintuitively, the initial conditions that we’re estimating are actually the conditions at the end of the assimilation cycle. This makes more sense when you think about it in terms of a cycle of “forecast” and “analysis” steps as seen in Figure 2. Essentially, the dynamics are contained entirely in the “forecast” step. The initial conditions we’re estimating are the conditions at the start of each forecast step.
>
> Using the numerical weather prediction example, you can imagine that every 12 hours the system may ingest temperature data recorded at certain points on the globe. The data assimilation process then takes in those observations, compares them with the predicted values and by balancing the observation noise and the uncertainty associated with the current forecast, comes up with a new estimate which is then pushed forward in time. The larger scale version of this is why the forecast for the weather this weekend may change dramatically over a few days - as new data is recorded. The same simulation may be producing the forecasts, but the assimilation process has updated the initial conditions used for the forecast so that the results may be notably different.
>
> This naturally leads to the question - “why are we not using forecast loss then if the forecast quality is what we actually care about?”. The reason here is actually related to Proposition 2 in our theoretical section which is that forecast error in chaotic systems is actually highly dependent on the system itself. While we can use one term to bound the other, the value of these bounds can vary depending on the system. As a result, the average RMSE between the analysis estimate and the true estimate across all observation points is actually fairly standard in data assimilation contexts. You can imagine that potentially any of these time points might be the point at which you want to initialize a forecast and so they are all reasonable places to estimate the error of the current best estimate.
>
> ## Independent Neural Networks
> This is definitely something we could be more explicit about. This is currently addressed within the section starting at line 216. The short answer is that the abstract describes the fully general procedure, ie, if you had the minimal amount of information available, how would you have to implement it. This is discussed in the first paragraph.
>
> The second paragraph, which you quote, is specifically referring to the implementation used in the experiments. The experiments use a fairly standard subsample+noise observation operator. This second approach demonstrates that if you have additional information (like the form and spatial distribution of the observation operator) that this can easily be incorporated into the self-supervised assimilation framework. Per your feedback, it definitely seems like it would be worthwhile to include the fully general formulation to the comparison table.
>
> ### Figures
>
> Figure 6 - this was a mistake in the submission - it should 100% have a legend. For reference, the three lines are:
>
> - Orange - Naive attempt at training with noisy targets (RMSE(predicted analysis value at time t, observation value at time t)
> - Green - Purely supervised training
> - Blue - Our proposed self-supervised training method
>
> The reason why this is intriguing is that it’s actually highly unexpected that self-supervised training outperformed supervised training and implies that there is a benefit to receiving signal back through the simulation even in the case where there is ground truth. These results are actually averaged over ten seeds - it explicitly says this about the experiments in 5.2. Originally that paragraph came earlier and was referring to all experiments, but we missed restating it for 5.1 once that was moved. This is something we've corrected. We also plan to add the deviation shading there to make the distribution clearer.
>
> On the rest of the figures, yes. We agree that these are too small and should be increased in size and rendered smaller than we initially expected. 3, 5 and 6 in particular have significant whitespace which we can use to increase the size without needing to remove any content.
>
> ## Other notes
>
> > Their core section about self-supervised assimilation could be a lot clearer. At one point the authors write "letting us use the framework from [16] to analyze the problem” without further explaining the framework … This makes it very arduous for the reader to understand what they do
>
> Unfortunately, this is specifically truncated for space purposes. In the text, we’re following the convention common to a number of deep learning papers where the motivation and high level concept for theoretical claims are covered in the text, but the meat of the proof itself is deferred to the appendix. In the line that you’re quoting, we’re really just drawing connections for readers who don’t want to explore the supplementary materials, but the proof itself actually stands independently of the reference as we re-derive the part of their results relevant to our claims before going into our additions for exactly the reason you point out here. This does however take a significant amount of page space and so it was banished to the appendix. We’re not huge fans of this convention, but it is somewhat necessary for conference papers, particularly those that mix theoretical and empirical results, and we hope that you'll forgive us here.
>
> > They claim:” (Line 251) All methods tested have explicit knowledge of the system dynamics. Due to the challenge of learning chaotic dynamics, we found that methods which rely on learning the dynamics were uncompetitive” I am not sure that is true ...
>
> This boils down to the fact that learned dynamics introduces an additional source of error. Even if you have a relatively strong learned dynamics model, a combined learned/DA system would have error introduced both by the dynamics and by the DA. If you look at the results, the error is actually quite low relative to the observation noise. Thus, for a learned dynamics approach to be competitive, it would have to be nearly exact. We are not presently aware of any reported results that achieve losses on the scale required on these systems while using noisy, incomplete, and sparsely sampled data to train. The method proposed by Brajard et al (reference 39) comes close, but is still at a magnitude where it would not be competitive against a priori known dynamics. Learned vs given dynamics is just not a fair comparison which is why we don't explore it.
>
> That's it for our rebuttal. If we were able to address any of your concerns leading to the current rating, we'd ask that you please consider increasing the score. If not, we still appreciate the feedback. Thank you.

---

> > ### Comment · Reviewer_NmPU · 2021-08-14
> > **Thanks for your response!**
> >
> > Thanks for the clarification! Most of my initial concerns are addressed. I have updated my score to reflect this.

---

### Decision · Program_Chairs · 2021-09-27

**Decision:**

Accept (Poster)

**Comment:**

From the SAC. This is an instance where the rebuttal and the discussion worked. While the original decision for this paper was to not accept, it is being raised to a recommended accept. The primary reason is the quality of the rebuttal, and the useful technical discussion between authors and reviewers that ensued that seems to have been revealing (in particular, reviewer kG8H). To the authors: I trust that you will take all reviewer feedback into account and most importantly, that all of the things in your rebuttal and discussion that were promised will be done in the next version of the paper.